# Applying Enhanced Real-Time Monitoring and Counting Method for Effective Traffic Management in Tashkent

**DOI:** 10.3390/s23115007

**Published:** 2023-05-23

**Authors:** Alpamis Kutlimuratov, Jamshid Khamzaev, Temur Kuchkorov, Muhammad Shahid Anwar, Ahyoung Choi

**Affiliations:** 1Department of AI, Software Gachon University, Seongnam-si 13120, Republic of Korea; 2Department of Information-Computer Technologies and Programming, Tashkent University of Information Technologies Named after Muhammad Al-Khwarizmi, Tashkent 100200, Uzbekistan; 3Department of Computer Systems, Tashkent University of Information Technologies Named after Muhammad Al-Khwarizmi, Tashkent 100200, Uzbekistan

**Keywords:** vehicle counting, YOLOv5, intelligent transportation system, smart city

## Abstract

This study describes an applied and enhanced real-time vehicle-counting system that is an integral part of intelligent transportation systems. The primary objective of this study was to develop an accurate and reliable real-time system for vehicle counting to mitigate traffic congestion in a designated area. The proposed system can identify and track objects inside the region of interest and count detected vehicles. To enhance the accuracy of the system, we used the You Only Look Once version 5 (YOLOv5) model for vehicle identification owing to its high performance and short computing time. Vehicle tracking and the number of vehicles acquired used the DeepSort algorithm with the Kalman filter and Mahalanobis distance as the main components of the algorithm and the proposed simulated loop technique, respectively. Empirical results were obtained using video images taken from a closed-circuit television (CCTV) camera on Tashkent roads and show that the counting system can produce 98.1% accuracy in 0.2408 s.

## 1. Introduction

Transportation is an essential part of daily life in today’s fast-paced world. According to recent statistics, it is anticipated that there will be approximately 1.45 billion vehicles across the globe by the end of 2023, with nearly 1.1 billion of those classified as passenger automobiles. Therefore, traffic issues in urban centers will almost certainly increase. Traffic congestion [1] is a major problem in many urban areas, causing delays, frustration, and decreased quality of life for commuters. To address this issue appropriately, several developed nations, including the United States, South Korea, and Japan [2,3], have begun to incorporate intelligent transportation systems (ITSs) [4,5,6]. ITSs have the potential to significantly reduce traffic congestion in urban areas by improving traffic flow, providing real-time information to drivers, and encouraging the use of alternative modes of transportation. By implementing these strategies, cities can improve the overall efficiency of their transportation systems, reduce congestion, and improve the quality of life for their residents. Effective traffic management requires accurate and timely data on traffic flow and volume, which can be obtained by counting and monitoring vehicles. Vehicle counting is an essential component of traffic management, particularly in the context of traffic congestion. Accurate vehicle counting allows transportation planners to understand traffic patterns and make informed decisions about traffic management strategies.

The primary components of an ITS are vehicle recognition and counting [7,8,9]. Researchers and engineers are creating intelligent traffic systems to enhance traffic light signal efficiency and reduce traffic congestion. Several scientific and experimental studies have been conducted to address these issues. In addition, an increase in data sources, such as video surveillance, allows for the efficient construction of vehicle-counting and monitoring systems. Recent developments in this field have led to new and more accurate methods for counting and monitoring vehicles on streets. For example, researchers have developed new machine learning algorithms [10,11,12] that can accurately detect and count vehicles in real-time, even in challenging environments. Other researchers have explored the use of crowdsourcing to collect real-time traffic data [13,14] from mobile devices, which can be used to inform traffic management decisions. Monitoring vehicles using traffic-surveillance videos comprises two parts: detection and counting. Deep learning object identification, frame difference, optical flow, and background removal [15,16] have been used for automobile recognition. However, these methods require large datasets and are challenging to implement using traffic surveillance videos. Tracking and detection regions are the main components of vehicle counting. For detection, a simulated detection region is created in the video to detect whether automobiles are moving in the respective region. It is difficult to maintain lanes or drive in a straight line; therefore, the vehicle frequently falls out of the region. By contrast, tracking identifies and counts using the direction of each automobile in each video frame. It is highly accurate but expensive to compute. Therefore, these aspects remain significant topics that must be investigated by researchers [17]. Despite deep learning and other advances, some limitations remain in the use of vehicle-counting and monitoring techniques for traffic management. For example, some methods may be expensive to implement on a large scale, whereas others may not be suitable in certain environments. In addition, the accuracy of some techniques may be affected by factors such as weather conditions, vehicle speed, and road layout. Overall, the reliability and time efficiency of vehicle monitoring, detection, and counting can be considerably enhanced by applying successful datasets and deep learning approaches.

Our study contributes to this field by exploring the use of a novel functional approach for vehicle counting and investigating its impact on traffic congestion. Therefore, in this study, we suggest using YOLOv5 and DeepSort as foundational approaches for vehicle detection and tracking, respectively. YOLOv5 has been optimized to reduce the likelihood of incorrect identification and to maximize its computational effectiveness. In order to optimize the efficiency of the YOLOv5-CSP model, we have made improvements to its Cross-Stage-Partial (CSP) structure. The CSP structure revolves around the concept of dividing the extracted features into two distinct sets. This division enables us to apply different processing stages to each set, thereby achieving a more refined representation of the input data. Subsequently, DeepSort, with its planned and efficient characteristics, was employed to solve the issue of tracking objects that were absent from certain frames because of complex backgrounds. Considering the aims of the system, we attempted to use a method called the simulated loop to count the number of cars on the roadways. The key contributions of this study are summarized as follows:A system has been developed that enables the accurate counting of numerous vehicles in several moving directions.The integration of YOLOv5 and modifications to the CSP structure have significantly enhanced the system’s accuracy, improving vehicle counting and detecting.A simulated loop technique has been introduced to avoid counting adjacent vehicles as a single unit in dense traffic scenarios.Through innovative modification to the CSP structure, we have streamlined the vehicle counting system, reducing the number of parameters and improving overall performance, processing times, and resource utilization.Through algorithmic enhancements and computational optimizations, we have achieved a significant reduction in vehicle counting time, resulting in improved system efficiency and enabling timely decision-making and analysis in traffic management and monitoring applications.

The remainder of this study is organized as follows. Recent developments and existing research related to vehicle counting and monitoring techniques are discussed in Section 2. In Section 3 and Section 4, we detail the proposed method and compare it with other methods through experimental evaluations. The conclusions of this study and their scope are outlined in Section 5. Overall, most of the selected references are relatively recent.

## 2. Related Work

In this section, we provide an overview of the current literature on the topic of vehicle counting and monitoring techniques, along with their potential impact on traffic congestion. In recent years, there has been a growing interest in using vehicle counting and monitoring systems to improve traffic flow, reduce congestion, and enhance overall transportation efficiency. As such, there is a vast body of literature available on this topic that explores various techniques, approaches, and systems for collecting and analyzing traffic data. The literature on vehicle counting and monitoring techniques covers a wide range of topics, including the use of video-based monitoring systems, radar-based systems, and inductive loop detectors. Various research studies have investigated the accuracy, reliability, and effectiveness of these systems in collecting and analyzing traffic data. Moreover, the literature also explores the different factors that can influence traffic flow, such as vehicle types, traffic patterns, and road conditions.

Object detection and tracking are the primary tasks of building a vehicle-counting model. Various methods exist for object detection and tracking, including traditional machine learning and deep learning methods. The authors of [18] combined support vector machine (SVM) and SIFT algorithms to improve vehicle detection accuracy by utilizing sliding windows and pooling techniques. In [19], a detection model for images that obtain car localization based on background subtraction and a Gaussian mixture model were proposed. Deep learning models, such as convolutional neural networks [7,17,20,21], have been shown to achieve high accuracy in object detection and tracking tasks and have been applied to vehicle counting and monitoring with promising results. Moreover, [22] developed a car recognition and tracking model using a joint probability method with radar and camera data. A strategy for detecting cars in aerial photos that only required a single stage and did not require the use of anchor points was proposed in [23]. Using a fully convolutional network to make direct predictions of high-level car attributes, the challenge of detecting vehicles can be converted into a multiplex subproblem. Furthermore, many researchers [8,24,25,26] have integrated YOLO [27] into various fields, including transport systems. YOLO can be used in ITSs to detect and localize vehicles in real-time video streams. Thus, YOLO can be found in recent studies [28,29,30], which constructed ITSs. A small deep network based on YOLOv3 was developed in [31] to enhance detection accuracy and speed by combining the spatial pooling technique in the network and applying it in a real-time environment.

Article [32] suggests a three-stage process that consists of object detection, object tracking, and trajectory processing to create a framework for counting vehicles using video, which can offer valuable information about traffic flow. The framework attempts to mitigate scene direction and manage difficult situations in videos. Another study [33] introduced an innovative system for detecting and tracking vehicles using visual input. The system comprises four primary stages: identifying the foreground, extracting relevant features, analyzing those features, and counting and tracking vehicles. However, the algorithms that have been created are not yet versatile, and their effectiveness relies heavily on the comprehensiveness of the training dataset and how well the dataset corresponds to normal operating circumstances. Furthermore, real-time processing and robustness to environmental factors such as weather conditions, camera vibrations, and changes in camera position remain challenging for researchers.

Overall, vehicle counting and monitoring are complex tasks that require the integration of multiple techniques and approaches. Although there is no universal solution, advances in technology and data analysis methods are expected to improve the accuracy and efficiency of vehicle-counting and monitoring systems. Therefore, in this study, we attempted to enhance the vehicle-counting procedure and apply it to Tashkent roads to obtain experimental evidence. The subsequent sections provide a comprehensive description of the entire process involved in the proposed system and describe the experimental results.

## 3. Methodology

This section provides a comprehensive explanation of the proposed system, designed specifically for counting vehicles in real-time scenarios on Tashkent roads. The system comprises distinct components, each of which contributes significantly to the accurate counting of vehicles. Figure 1 illustrates the complete implementation process. YOLOv5 has undergone extensive and meticulous optimization not only to lessen the possibility of inaccurate identification but also to increase its computational efficiency. Subsequently, DeepSort was used to handle the problem of tracking items that were missing from specific frames owing to complicated backdrops. DeepSort is equipped with well-planned and effective features, which it uses to solve the problem. Considering the goals of the system, we applied a technique known as the simulated loop to count the number of vehicles driving on the roads. The following sections provide a more comprehensive description of each element in the proposed system.

### 3.1. Vehicle Detection Model

We used YOLOv5 as the baseline model because it was the most appropriate for both data collection and our research goals. Additionally, the newly released convolutional neural network in YOLOv5 can detect both stationary and moving vehicles with outstanding speed and precision in real-time. Determining traffic flow on highways requires a high level of detection precision speed, and the small size of the model affects the effectiveness of its inferences in resource-constrained embedded systems. Our research also focuses on enhancing the capabilities of the CSP (Cross-Stage Partial) model [34], particularly in extracting relevant and valuable characteristics from input video frames. In this enhanced version, we have integrated the powerful YOLOv5 network as the backbone, further elevating the model’s efficiency and performance. To achieve these improvements, we have introduced gradient modifications within the feature map. This modification addresses a common issue in complex networks where redundant gradient data can lead to inefficiencies. By leveraging gradient modifications as part of the feature map, we effectively eliminate this issue and optimize the model’s computational efficiency. By minimizing redundant gradient data, the CSP model significantly reduces the number of hyperparameters and floating-point operations per second. This has a twofold effect: it increases the prediction speed and precision while simultaneously decreasing the overall size of the model. This improvement is crucial in real-time vehicle counting applications, where fast and accurate predictions are essential. In our improved version, we have fine-tuned the splitting ratio based on extensive experimentation. Figure 2 depicts the splitting of features into two separate sets. The first undergoes extra stages of improvement, whereas the other bypasses them. In most cases, the features of the input will be divided in half (γ=0.5). However, we drop to 0.25 to accelerate the system by reducing the number of parameters, which in turn boosts the frames per second. Nonetheless, this will decrease the mean average precision (mAP). Simultaneous batch normalization [35] is used for system pre-training to compensate for the drop in the mAP.

### 3.2. Multi-Vehicle Tracking

Multi-vehicle tracking is another crucial function of the proposed system. In this system, the DeepSort algorithm [36] is used for online vehicle tracking because it enhances the vehicle detection process and flipping between vehicles. For tracking, we take a broad approach and suppose that we have no information about the movement of the camera in relation to a static background and that the CCTV camera is positioned.

During the operation of the DeepSort algorithm, objects exhibit two characteristics: movement dynamics and appearance. For dynamics, parameters, such as the height, aspect ratio, region of interest (ROI), and scene coordinates, are typically filtered and predicted using a Kalman filter. Accordingly, these parameters form the basis of the tracking case. We used a Kalman filter with a steady-speed motion model and a linear observation model. In this model, we accepted ROI coordinates as observational data (appearance) of the vehicle state. It is possible to use a pre-trained neural network with the structure shown in Table 1 for observational data.

The Mahalanobis distance (Equation (1)) between the estimated Kalman states and the most recent observational data was used to include the motion data.
(1)$$D_{M}(\vec{x})=\sqrt{\left(\vec{x}-\vec{\mu }\right){}^{T}S^{-1}(\vec{x}-\vec{\mu })}$$

Here,

The difference is the difference between measurement space and ROI;S represents a covariance matrix.

The Mahalanobis distance estimates the error at the state level by calculating the number of standard deviations by which the detection deviates from the average track position.

### 3.3. Vehicle Counting

In most cases, modern real-time vehicle-counting systems use counting techniques regardless of road conditions or traffic situations. For instance, traffic congestion increases the likelihood of incorrectly counting several cars as a single unit because the cars are crowded together and moving slowly. Methods that rely on line detection are well-suited for accurately counting vehicles moving at high speeds, but they may struggle in congested traffic where vehicles are closely spaced and moving slowly, leading to the risk of counting adjacent vehicles as a single unit. We assume that to overcome this challenge, a simulated loop, which can be seen as an extension of parallel line detection or simulations of traditional inductance loops, has been introduced. Therefore, our system utilizes the simulated loop technique to improve counting accuracy in dense traffic scenarios, offering valuable insights for traffic management and analysis. In our case, the simulated loop is an ROI that includes all road lanes. Counting in heavy traffic can be performed efficiently using simulated loop approaches [37,38]. In such scenarios, multiple vehicles may be closely packed together, moving slowly, or experiencing temporary stops. These conditions can make it difficult to count individual vehicles using traditional methods accurately. By including all lanes within the ROI, the method ensures that no vehicles are missed during the counting process. This comprehensive approach eliminates the need for separate counting of individual lanes, reducing complexity and increasing the accuracy and efficiency of the counting process. Moreover, the method utilizes the YOLOv5 algorithm to detect and identify vehicles within the frames accurately. YOLOv5 is designed to handle challenging conditions, including variations in lighting, weather, and object appearances, to ensure reliable vehicle detection regardless of weather conditions.

There is an ROI on the road, and its length is identical to that of the road. This is a made-up ROI into which users can input regional parameters.

Each ROI in the frames is assigned a progress indicator expressed by $F_{pi}$ and formulated as in Equation (2).
(2)$$F_{pi}=\left\{\begin{array}{c} 0,\;when\;ROI\;is\;clear\\ 1,\;otherwise \end{array}\right.$$

Calculating the ratio of the detected vehicle pixels to the total number of pixels and the average width of the vehicle in the ROI is a concrete method to obtain the value of $F_{pi}$.

First, we computed the binary frame with vehicles spotted in the ROI. Here, we determined the number of pixels expressed by P that constitute a vehicle. The following step involved assuming that the dimensions of the ROI are a×b, where *a* and *b* refer to the length and width of the ROI, respectively, and have a pixel count. Equation (3) can be used to determine the vehicle pixel ratio, which is denoted by the symbol μ.
(3)$$\mu =\frac{P}{a\times b}$$

The empirical findings demonstrate that $F_{pi}$ can be expressed in the following format.
(4)$$F_{pi}=\left\{\begin{array}{c} 0,\;otherwise\\ 1,\;\mathrm{if}\mu \geq 0.1\;\mathrm{and}\;\rho \geq 0.35 \end{array}\right.$$
where ρ is the ratio of the width of the vehicle to the width of the ROI. When $C_{i}$ stands for the current number of the *i*-th lane of the road, this number changes as follows.
(5)$$C_{i}=\left\{\begin{array}{c} C_{i};F_{pi}:0\to 0\\ C_{i}+1;F_{pi}:0\to 1\\ C_{i};F_{pi}:1\to 0\\ C_{i};F_{pi}:1\to 1 \end{array}\right.$$

## 4. Experiment

### 4.1. Hardware and Software Configurations

The proposed model was implemented and tested using a core i9-13900K CPU, 24 GB GPU, and 64 GB RAM. In addition, real-time videos were captured by a CCTV camera. Table 2 lists the hardware and software configurations used to develop the proposed model.

### 4.2. Evaluation Metric

During the evaluation of network performance, the average precision (AP) is the primary metric used for training, and the performance of the trained network is measured using the validation set. The expressions for P (Precision) and R (Recall) are as follows:(6)$$\left\{\begin{array}{c} Recall=TP/(TP+FN)\\ Precision=TP/(TP+FP) \end{array}\right.$$

The True Positives (TP) are the samples that are truly positive and are correctly classified as positive by the classifier. The True Negatives (TN) are the samples that are truly negative and are correctly classified as negative by the classifier. False Positives (FP) are the samples that are actually negative but are incorrectly classified as positive by the classifier. False Negatives (FN) are the samples that are actually positive but are incorrectly classified as negative by the classifier.

The AP is the area enclosed by the Precision–Recall (P–R) curve, which is used to evaluate the performance of a classifier. Typically, a higher AP value indicates a better classifier. The AP is the average of the AP values for each category, representing a composite measure of the average precision of the detected targets across all categories.

### 4.3. Dataset

In order to facilitate our research and enhance the accuracy of our system, we have meticulously constructed a new dataset using data collected from CCTV cameras. In the experiment, we utilized day and night CCTV records of Tashkent streets divided into 4 and 5 min videos with 1440 × 1080 resolution. Videos were captured using a static CCTV camera installed at a height of 15 m. In order to effectively evaluate the performance of our vehicle counting system, 1 h of video footage was divided into 5 × 4 min and 5 × 5 min video chunks (datasets). The first 5 × 4 min and 2 × 5 min videos were used for training, while the last 3 × 5 min videos were for testing purposes. This dataset served as the foundation for conducting our experiments and obtaining our test results. As well this dataset serves as a valuable resource for training and evaluating our vehicle counting algorithms, providing a comprehensive representation of real-world scenarios and diverse traffic conditions.

Information regarding the dataset is presented in Table 3, and sample images are visually represented in Figure 3.

### 4.4. Results

The developed system counted vehicles during the daytime under normal and during nighttime under rainy weather conditions. The tests were conducted with a moving background, making it difficult to detect and count vehicles. Figure 4 and Table 4 and Table 5 present the obtained results and their comparisons with existing methods.

As shown in Table 4, the tests were conducted using 10 test videos. The system counted vehicles in both forward and backward directions in moving background situations. The illustrated accuracy was the average accuracy of the three vehicle types in each test case. The movement of trucks on Tashkent roads is prohibited at night. Thus, the proposed system counted the number of trucks as zero in the nighttime test cases. Furthermore, Table 5 shows a comparison of the proposed system with other systems in terms of time and accuracy. The accuracy indicated in Table 5 is the average accuracy of ten test cases conducted using the proposed system. The following methods were selected as benchmarks for comparison with the proposed system.

Yolo4-CSP [19]: a detection-tracking-counting method for movement-specific vehicles. The CSP architecture divides the backbone network into two branches: a main branch and a cross-connection branch. The main branch is responsible for feature extraction, while the cross-connection branch is used to transmit features across different stages of the network. This cross-connection allows for better information flow and reduces the loss of feature information during forward propagation.VC-UAV [7]: a multi-object management module capable of effectively analyzing and validating the status of tracked vehicles through multithreading. The system utilizes a visual serving approach to track a target object, allowing the UAV to move in real-time to keep the object in view. The system is composed of several components, including a camera for image acquisition, an onboard computer for processing and control, and a set of motors for maneuvering the UAV.VH-CMT [18]: a correlation-matched multi-vehicle tracking and vehicle-counting approach. It utilizes both appearance and motion information to improve the accuracy of object detection and tracking. One of the key characteristics of the VH-CMT model is its ability to use contextual information to aid in object detection and tracking. Specifically, the model takes into account the motion trajectories of other objects in the scene, as well as the spatial relationships between them, to improve the accuracy of object detection and tracking.

As shown in Table 5, the proposed system is superior to the alternatives in terms of both speed and precision. In particular, even a marginal increase in speed with real-time systems allows earlier detection and counting, which may aid in managing traffic flows at subsequent crossings. The system delivers its results in 0.2408 s, whereas the nearest alternative, Yolo-CSP, takes 0.249 s. The overall system accuracy is 98.10%, whereas Yolo4-CSP, VC-UAV, and VH-CMT averaged 94.76%, 95.54%, and 93.11%, respectively. To further enhance the performance of the YOLOv5-CSP-0.25 model, we conducted a pre-training phase with synchronized batch normalization. By incorporating synchronized batch normalization into the training process, we aimed to improve the average accuracy metric, which is a crucial indicator of detection accuracy. This pre-trained variant of the model denoted as YOLOv5-CSP-0.25-sync, was trained using a batch size of 8, ensuring efficient utilization of computational resources while maintaining high performance. By leveraging the effectiveness of the YOLOv5-CSP-0.25 architecture and augmenting it with synchronized batch normalization during pre-training, we have achieved a refined model that strikes a balance between accuracy and efficiency, thereby enhancing the overall performance of our vehicle counting system. Our study includes an evaluation of our system’s effectiveness. TensorFlow is a framework for deep learning that we utilized to train and evaluate our model’s prediction accuracy. Our findings demonstrate a high degree of accuracy, with a 98.1% rate of correct predictions. This demonstrates the reliability of our method for predicting outcomes from input variables.

We checked the accuracy functions of the visual representations from both the training and validation sets to ensure the network was trained without overfitting (Figure 5). The findings show that the network was trained efficiently without overfitting, which is essential for the system’s robustness and precision. This result proves that our method is capable of producing reliable output predictions and illustrates its usefulness in the real world.

### 4.5. Discussion

This study offers a significant approach to easing traffic congestion in the city of Tashkent. Congestion in urban areas may be alleviated by the use of real-time car counting systems that are both precise and dependable. Many important advances in vehicle counts and traffic congestion are made in this work. The unique functional technique we investigated relied on YOLOv5 for vehicle recognition and DeepSort for tracking. These methods were fine-tuned to improve the system’s precision while decreasing its computing burden. The system’s capacity to track a large number of cars traveling in different directions was greatly expanded by the use of the simulated loop approach for vehicle counting.

This study has made a number of important contributions, one of the most important being the establishment of a new dataset that is based on CCTV cameras. This dataset may be used to assess the accuracy and effectiveness of vehicle counting systems. Researchers in the area will have access to a significant resource in the form of this dataset, which will enable them to evaluate various methodologies and approaches and further enhance the accuracy and efficiency of vehicle counting systems. We also made significant improvements to the system’s accuracy and efficiency by reducing the number of parameters by making adjustments to the SCP structure and utilizing simultaneous batch normalization to control mean average precision. Several changes were made to improve the system’s overall effectiveness (mAP). The system’s efficacy in decreasing the negative effects of traffic congestion was improved not just by enhancing the accuracy but also by lowering the time required to count cars. By performing speed calculations, it is possible to ascertain the pace of the vehicle tracking algorithm that relies on the features of the proposed system. The system’s processing time increases as the number of cars increases in a given environment. Namely, when there are more vehicles present, additional features must be extracted, which increases the processing duration of the system. Despite the possibility of an increase in processing time, the vehicle counting system described in this study can still be regarded as nearly real-time. This indicates that the algorithm employed by this system is efficient enough to process multiple vehicles concurrently without significant delays or interruptions. Consequently, the proposed vehicle counting system could be an excellent solution for real-time applications requiring precise vehicle tracing.

This study’s findings demonstrate that the developed system can accurately count cars in real-time, with an accuracy of 98.1% in 0.2408 s. This degree of accuracy is amazing, demonstrating the power of deep learning algorithms in traffic control. This indicates that the system’s ability to identify and monitor vehicles in a given scene is extremely precise. In contrast, other systems, including Yolo4-CSP, VC-UAV, and VH-CMT, have lower accuracy rates than the system under discussion. Yolo4-CSP’s average accuracy is 94.76%, VC-UAV’s average accuracy is 95.54%, and VH-CMT’s average accuracy is 93.11%. According to the comparison of various systems, the system under consideration beats other current vehicle tracking systems. This might be ascribed to the suggested system’s vehicle tracking algorithm, which has shown to be very successful in identifying and tracking automobiles in real-time. Notwithstanding, it is vital to realize that the system’s performance may vary depending on environmental circumstances such as weather and lighting. This study also works on the development of a new dataset based on CCTV cameras. However, it does not elaborate on the diversity and representativeness of the dataset. The lack of information about the dataset raises concerns about the generalizability of the proposed approach to different real-world scenarios and camera setups. As well, the proposed system does not focus on the scalability of the proposed approach. It is crucial to assess whether the functional approach can handle large-scale scenarios with heavy traffic and multiple cameras. Scalability is essential to ensure the system can handle real-world traffic conditions effectively.

In addition, the system’s high accuracy rate may be useful in many contexts, including those where precision is paramount, such as traffic control, surveillance, and autonomous vehicle navigation.

The research described in this study constitutes a major accomplishment in the area of intelligent transportation systems and offers a realistic solution to the issue of excessive traffic congestion. Real-time automobile counting might provide valuable data to city planners and transportation authorities, enabling them to make educated choices regarding traffic management and road infrastructure expansion. This study lays the groundwork for future research and development in this field while also proving the use of deep learning algorithms and real-time vehicle-counting systems in minimizing the consequences of traffic congestion.

## 5. Conclusions and Future Scope

In this study, we propose an enhanced and applied vehicle-counting system in a moving background scenario based on advanced technologies to improve accuracy, reduce counting time, and manage traffic congestion on Tashkent roads. The proposed system is a component of the ITS of the Tashkent Smart City project and can be applied in different weather conditions, such as rain, snow, and wind. Specifically, the system can simultaneously count numerous moving vehicles to alleviate traffic congestion, manage traffic flow, and increase the effectiveness of traffic signals.

One of the contributions of the proposed system is that it reduces the counting time, which is a crucial aspect of real-time systems. Thus, early detection and counting can help manage traffic flows at subsequent intersections. Furthermore, we obtained a dataset for the use and evaluation of further research models. We analyzed the effectiveness of other deep learning approaches to verify the accuracy of our system further. The results of our experiments demonstrate the superior efficiency and accuracy of our vehicle-counting system. Future work will include the development of new algorithms for counting vehicles and traffic congestion using this system. Moreover, we aim to integrate a recommendation system [39,40,41] into the transport system to provide personalized and efficient travel options for drivers. Enhancing the overall user experience and improving the safety and efficiency of transportation using speech recognition models [42] will also be a future research domain.

Overall, we hope that the proposed system will play a crucial role in the ITS of any smart city for managing traffic flows and monitoring traffic congestion.

## Figures and Tables

**Figure 1 sensors-23-05007-f001:**
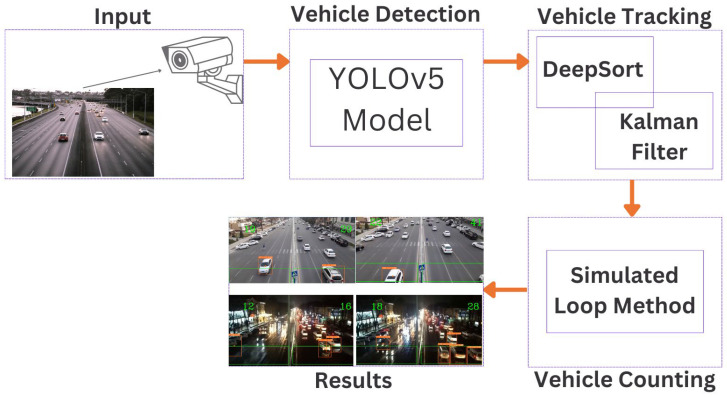
System workflow.

**Figure 2 sensors-23-05007-f002:**
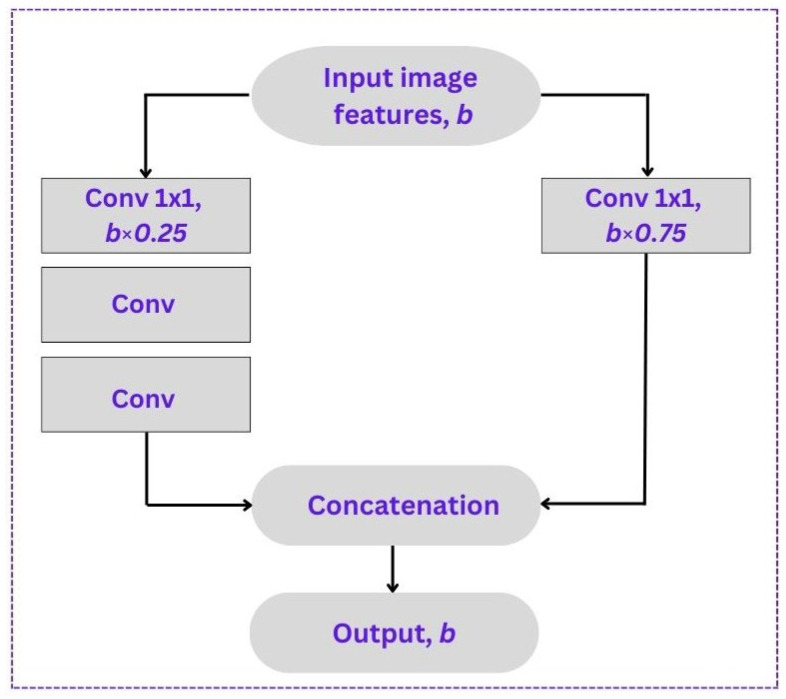
The CSP structure for input features.

**Figure 3 sensors-23-05007-f003:**
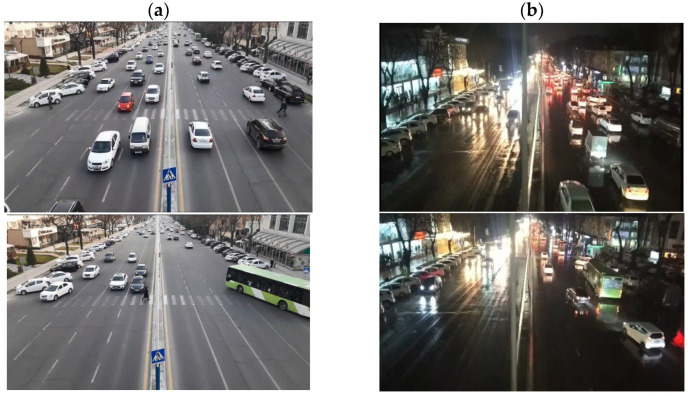
Data samples: (**a**) daytime and (**b**) nighttime.

**Figure 4 sensors-23-05007-f004:**
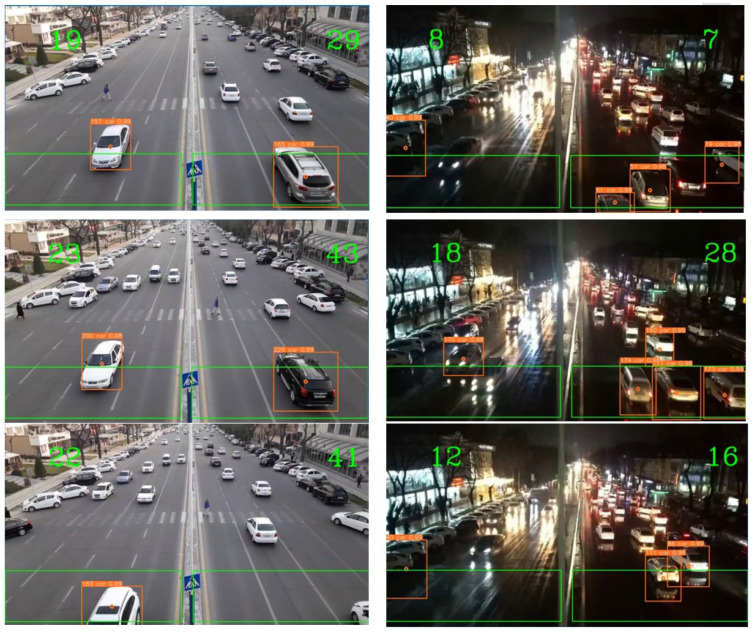
Examples of vehicle counting during daytime and nighttime.

**Figure 5 sensors-23-05007-f005:**
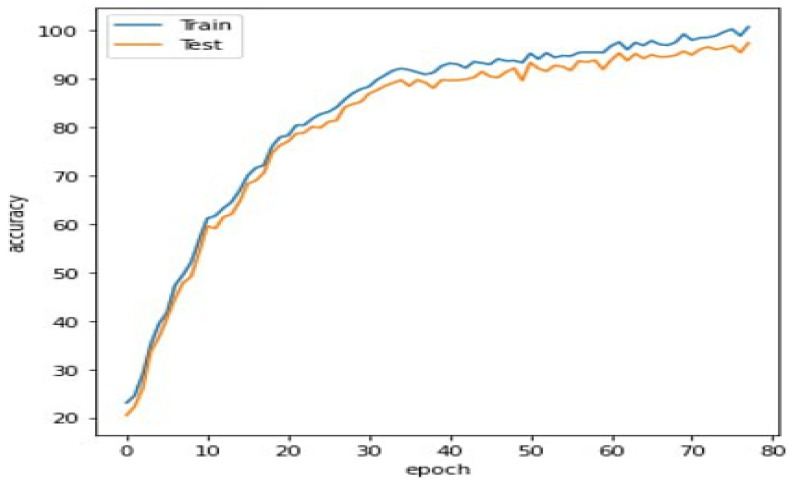
Evaluation of our system.

**Table 1 sensors-23-05007-t001:** Neural network parameters for observational data.

Name	Patch Size/Stride	Output Size
Conv 1	3 × 3/1	32 × 128 × 64
Conv 2	3 × 3/1	32 × 128 × 64
Max Pool 3	3 × 3/2	32 × 64 × 32
Residual 4	3 × 3/1	32 × 64 × 32
Residual 5	3 × 3/1	32 × 64 × 32
Residual 6	3 × 3/2	64 × 32 × 16
Residual 7	3 × 3/1	64 × 32 × 16
Residual 8	3 × 3/2	128 × 16 × 8
Residual 9	3 × 3/1	128 × 16 × 8
Dense 10	-	128
Batch and $\ell_{2}$ normalization		

**Table 2 sensors-23-05007-t002:** Hardware and software specifications.

	Hardware/Software	Configuration
CCTV (Input Data)	Smart HDx camera	1440 × 1080, 3.3×, Day/night camera
	Network connectivity	IEEE 802.3af, IEEE 802.3at/PoE Plus
	Power	24 V AC
Model Implementation	RAM	DDR4 64 GB
	GPU	GeForce RTX 3090 Ti, 24 GB GDDR6X, 384-bit
	CPU	Intel core i9-13900K
	Memory	SSD 1024 GB
	OS	Ubuntu
	Programming environment	Python, Anaconda, OpenCV, Pandas, YoLOv5, TensorFlow

**Table 3 sensors-23-05007-t003:** Dataset information.

Name	Length (min)	Background	Light	Directions
Video_data_1	4	Moving	Day	Forward/Backward
Video_data_2	4	Moving	Night	Forward/Backward
Video_data_3	4	Moving	Day	Forward/Backward
Video_data_4	4	Moving	Night	Forward/Backward
Video_data_5	4	Moving	Day	Forward/Backward
Video_data_6	5	Moving	Night	Forward/Backward
Video_data_7	5	Moving	Day	Forward/Backward
Video_data_8	5	Moving	Night	Forward/Backward
Video_data_9	5	Moving	Day	Forward/Backward
Video_data_10	5	Moving	Night	Forward/Backward

**Table 4 sensors-23-05007-t004:** Average accuracy results on counted vehicle types.

Test	All Vehicle Numbers	Counted Vehicle Types and Their Count			Directions	Accuracy (%)
		Bus	Car	Truck		
Test_1	226	46	173	3	Forward/Backward	98.2
Test_2	67	8	58	0	Forward/Backward	97.0
Test_3	285	63	214	2	Forward/Backward	97.8
Test_4	78	12	65	0	Forward/Backward	98.7
Test_5	261	50	200	6	Forward/Backward	98.0
Test_6	113	19	93	0	Forward/Backward	99.1
Test_7	317	71	233	6	Forward/Backward	97.79
Test_8	104	15	88	0	Forward/Backward	99.0
Test_9	342	79	244	11	Forward/Backward	97.6
Test_10	98	12	84	0	Forward/Backward	97.9

**Table 5 sensors-23-05007-t005:** Comparison of time and accuracy performances.

Model	Average Time (s)	Average Accuracy
Yolo4-CSP	0.249	94.76
VC-UAV	0.2712	95.54
VH-CMT	0.256	93.1
Proposed	0.2408	98.10

## Data Availability

Not applicable.

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
