# Peer review of "Applying Enhanced Real-Time Monitoring and Counting Method for Effective Traffic Management in Tashkent"

_sensors, 2023, doi:10.3390/s23115007_

Round 1

Reviewer 1 Report

Comments and suggestions for Authors

In this manuscript, the authors develop a real-time monitoring and counting system for traffic management. I’m impressed by the detailed work. There are several suggestions can help authors further improve the quality of the manuscript.

1.        The authors should pay a more attention to highlight the innovation of this work. Although the manuscript provides a systematic description about the vehicle counting system, the authors ignore to point out the novelty work within the system. Those methods and algorithms like YOLOv5 and DeepSort that authors employed are all mature technology and widely used in object detection task. As a journal paper, the innovation is the most important index to evaluate the valuable of the work. Therefore, the authors are suggested to supplement the main contribution of the work in the end of Introduction part and reorganized the content of Methodology part to present the innovation in method or algorithm.

2.        The CSP structure is relatively new to me. Is it a novelty work in this manuscript? If it is, I suggest the authors provide more description about the principle and mechanism within YOLOv5 framework.

3.        The entire structure or architecture of the model need to present in the Methodology part. Strictly speaking, figure 3 is not the illustration about the model structure. Those parameter configurations should be listed in Table for presentation.

4.        The authors do not provide the necessary description about the sample information in training set and testing set. Is it meet the requirement of independent and identically distributed? The authors need to clarify about it.

5.        The evaluation index of proposed method needs to be elaborated clearly. More specifically, what is the formula for accuracy? In this task, is the counting accuracy representing the ratio between detecting vehicle and total vehicle during period? Or the statistical accuracy into specific categories including bus, car and truck. Whether the designed system has function for vehicle categories recognition during objection detection? Or just foreground and background detection? The author should be clear about it.

6.        In comparison experiment, the authors need to briefly describe comparison approaches and provide corresponding parameter configuration to ensure that the comparison model is at the same level as proposed model for fair comparison. Meanwhile, the computational complexity comparison may be a good option for better verifying the effectiveness of proposed method.

7.        Overall, the quality of the schematic diagram provided in the manuscript is not up to the mark. Please provide high resolution figures.

1.        There are some typos and grammatical errors within the manuscript that need to be addressed.

Reviewer 2 Report

         In order to solve the problem of real-time monitoring and counting method for effective traffic management in Tashkent, the author used the YOLOv5 model for vehicle identification owing to its high performance and short computing time. However, there are still some problems in the manuscript, such as grammar problems, the lack of innovative algorithms. Furthermore, the literature review did not give a comprehensive discussion for counting method, thus, the introduction section needs some improvements.

1) In the " Abstract.", for the first occurrence of these abbreviations " YOLOv5, CCTV ", it needs to list the full name.

2)  In the " Keywords:", the first keyword "counting;" is changed to " Traffic Count".

3) The content after "in" is missing, on line 118.

(Please see the picture in pdf file)

4) In "2. Related Work", the introduction of the author's work is only one sentence, which is not comprehensive enough and needs to be supplemented.

5) The words and picture are deformation in " Figure 1. System workflow". The block diagram needs to be modified to be more aesthetically pleasing.

(Please see the picture in pdf file)

6) The abbreviation that appears for the first time in the paper needs to be written in full, such as CSP, mAP, on lines 159 and 169.

(Please see the picture in pdf file)

7) The label is repeated, on lines 174 and 198

(Please see the picture in pdf file)

8) In "3. Methods", the description of research methods is not detailed enough, and the repeatability of the research needs to be improved.

9) The inadequacies of the system are not specified in "4.4 Discussion".

10) Before the "References", some additional information of the paper was not perfected, such as Institutional Review Board Statement, Informed Consent Statement, Data Availability Statement.

11) On line 239, the statement is incorrect.                      

 ï¼ˆPlease see the picture in pdf file)

12) There are some minor errors in the format of the references, and the authors need to write the correct format according to different types of documents.

13) This sentence is in the wrong tense. It should be in the past tense.

(Please see the picture in pdf file)

There are still some problems in the manuscript, such as grammar problems.  There are some minor errors in the format of the references, and the authors need to write the correct format according to different types of documents.

Reviewer 3 Report

This paper presents a real-time vehicle-counting method using the DeepSort algorithm with the Kalman filter for intelligent transport systems. The introduction states the main purpose of the paper, and the relation between the paper and the previous works is clearly explained. The performance of the presented counting method is illustrated through experiments.

The following are my specific comments:

 (1)  The full names of the abbreviations should be given when they firstly appear in the paper, such as “YOLOv5”, “CCTV”, “SVM” and “CSP”.

(2)  How are problems caused with the weather conditions and camera vibrations coped with in the presented counting method? It is better to provide some explanations.

(3)  For clarity, the authors are suggested to provide a pseudo code for the presented counting method.

(4)  The mathematics model for the design of the Kalman filter should be given in Section 3. How is the covariance matrix S calculated? It is better to provide some explanations.

(5)  It seems that a symbol in equation (4) is not defined. Please check.

(6)  It is stated in Section 3 that traffic congestion increases the likelihood of incorrectly counting several cars as a single unit. It is better to illustrate the performance of the presented method in the case of traffic congestion.

(7)  Why is the presented method superior to the Yolo4-CSP, VC-UAV and VH-CMT in term of precision? Try to provide some remarks.

(8)  It is stated in conclusion that the presented method can be applied in different weather conditions. However, this result is not verified in the experiment.

Round 2

Reviewer 1 Report

Comments and suggestions for Authors

The authors of the manuscript titled " Applying enhanced real-time monitoring and counting method for effective traffic management in Tashkent" have satisfactorily addressed the majority of reviewer concerns, significantly enhancing the paper's quality. There are some small suggestions can make the paper of interest and benefit to a wider cross-section of readers.

1.       The main contribution work that the authors supplemented in the end of introduction are suggested to be summarized. The aim of this presentation is to point out the most novelty work in the method, algorithm or system such as model structure, optimization procedure and system efficiency. The description about dataset and functional effectiveness can be moved to the experiment part instead of introduction part.

2.       Some figures such as Figure 1,2,5 is still unclear and difficult to read. Please enhance the quality and legibility of these figures.

Reviewer 2 Report

The author has revised most of the questions raised last time.

It is hoped the author can read the manuscript thoroughly and detect errors in the speech and text.

Reviewer 3 Report

I believe that the manuscript has been improved and now warrants publication in Sensors.
